# Inhibition of Calcineurin with FK506 Reduces Tau Levels and Attenuates Synaptic Impairment Driven by Tau Oligomers in the Hippocampus of Male Mouse Models

**DOI:** 10.3390/ijms25169092

**Published:** 2024-08-22

**Authors:** Michela Marcatti, Batbayar Tumurbaatar, Michela Borghi, Jutatip Guptarak, Wen-Ru Zhang, Balaji Krishnan, Rakez Kayed, Anna Fracassi, Giulio Taglialatela

**Affiliations:** Mitchell Center for Neurodegenerative Disease, Department of Neurology, The University of Texas Medical Branch at Galveston, Galveston, TX 77550, USA; mimarcat@utmb.edu (M.M.); batumurb@utmb.edu (B.T.); mborghi@uni-mainz.de (M.B.); juguptat@utmb.edu (J.G.); wezhang@utmb.edu (W.-R.Z.); bakrishn@utmb.edu (B.K.); rakayed@utmb.edu (R.K.)

**Keywords:** Tau, calcineurin, FK506, Alzheimer’s disease, oligomers

## Abstract

Alzheimer’s disease (AD) is the most common age-associated neurodegenerative disorder, characterized by progressive cognitive decline, memory impairment, and structural brain changes, primarily involving Aβ plaques and neurofibrillary tangles of hyperphosphorylated tau protein. Recent research highlights the significance of smaller Aβ and Tau oligomeric aggregates (AβO and TauO, respectively) in synaptic dysfunction and disease progression. Calcineurin (CaN), a key calcium/calmodulin-dependent player in regulating synaptic function in the central nervous system (CNS) is implicated in mediating detrimental effects of AβO on synapses and memory function in AD. This study aims to investigate the specific impact of CaN on both exogenous and endogenous TauO through the acute and chronic inhibition of CaN. We previously demonstrated the protective effect against AD of the immunosuppressant CaN inhibitor, FK506, but its influence on TauO remains unclear. In this study, we explored the short-term effects of acute CaN inhibition on TauO phosphorylation and TauO-induced memory deficits and synaptic dysfunction. Mice received FK506 post-TauO intracerebroventricular injection and TauO levels and phosphorylation were assessed, examining their impact on CaN and GSK-3β. The study investigated FK506 preventive/reversal effects on TauO-induced clustering of CaN and GSK-3β. Memory and synaptic function in TauO-injected mice were evaluated with/without FK506. Chronic FK506 treatment in 3xTgAD mice explored its influence on CaN, Aβ, and Tau levels. This study underscores the significant influence of CaN inhibition on TauO and associated AD pathology, suggesting therapeutic potential in targeting CaN for addressing various aspects of AD onset and progression. These findings provide valuable insights for potential interventions in AD, emphasizing the need for further exploration of CaN-targeted strategies.

## 1. Introduction

Alzheimer’s disease (AD) is a progressive neurodegenerative disorder and the most common form of age-related dementia, currently affecting more than 6 million in the Unites States alone [1]. The histopathological features of AD involve the formation of extracellular plaques containing insoluble fibrillar aggregates of the amyloid beta peptide (Aβ) and intracellular neurofibrillary tangles of hyperphosphorylated Tau protein. These large aggregates are formed through a process of protein misfolding, aggregation, and deposition that begins with the formation of small soluble oligomers [2,3,4,5]. These soluble oligomeric forms of Aβ (AβO) and Tau (TauO) have been identified as the most neurotoxic species involved in synaptic degeneration [5,6,7]. In vitro and in vivo experiments aimed at studying synaptic dysfunctions in AD revealed that these oligomers reduce long-term potentiation (LTP) and enhance long-term depression (LTD) by altering Ca^2+^ signaling, leading to disruption of Ca^2+^ homeostasis, which is fundamental for proper synaptic functioning [8,9,10,11,12,13,14]. Although the exact mechanisms by which oligomers promote neuronal degeneration are not fully understood, regulation of the Ca^2+^/calmodulin-dependent protein phosphatase calcineurin (CaN) seems to play a key role [15]. Indeed, CaN is abundantly expressed in neurons, where it regulates the processes of synaptic plasticity, learning and memory, and neuronal apoptosis. Increased CaN activity has been found following acute treatment with both Aβ and α-synuclein (α-syn) oligomers [16,17,18]. Moreover, treatment with CaN activity inhibitor FK506/Tacrolimus, an FDA-approved immunosuppressive drug, restored memory and synaptic alterations caused by both Aβ and α-syn oligomers in in vivo and ex vivo behavioral and electrophysiological studies [17,19,20,21]. Evidence linking other oligomeric species to CaN dysregulation suggests that TauO could also induce neuronal dysfunctions, leading to early cognitive impairment in a CaN-dependent fashion. Notably, we reported that humans chronically treated with the CaN activity inhibitor FK506 were protected from developing AD, suggesting a key role of CaN in onset and progression of AD [22,23]. However, the impact of CaN on TauO remains unresolved. Based on this evidence, in the present study, we investigated the effects of CaN activity inhibition by FK506 on TauO deposition, phosphorylation, and synaptotoxicity. Our findings suggest that inhibiting CaN activity can effectively counteract TauO toxicity and promote their clearance by reinstating an efficient phagocytic activation of the autophagic pathway. Therefore, treatment with FK506 may hold promising therapeutic potential in mitigating AD pathology.

## 2. Results

### 2.1. Modulation of TauO-Induced Changes in CaN Protein Levels by FK506 in Mouse Hippocampus

To understand how TauO affect CaN protein levels in the hippocampus, we ICV injected preformed recombinant TauO [24] into the brains of 12 weeks old C57BL/6J male wild-type mice. Mice were further treated with the CaN activity inhibitor FK506 18 h post ICV injection of TauO and sacrificed 6 h later, as described in the methods section, and illustrated in Figure 1. We evaluated CaN levels by Western blotting and immunofluorescence analyses across all experimental groups focusing on the hippocampus, given its key role in memory processing and its early susceptibility to pathological changes in AD (Braak stages I and II) [25]. Although Western blot analyses did not show significant differences in CaN protein levels across the experimental groups (Figure 2a), immunofluorescence analyses revealed a significant increase in CaN levels in all three analyzed hippocampal regions—Dentate Gyrus (DG), CA1, and CA3—upon TauO injection (Figure 2b–d).

Such increased CaN was particularly evident in the cell soma, reflecting the arrangement of neuronal cell bodies in the hippocampus. In the DG (Figure 2b), CaN was mostly localized in the soma of granule cells neurons (white arrows) and in the hilus (white arrowheads). In the CA1 and CA3, CaN distribution was mostly prevalent in the soma, but it was also prevalent in the processes of pyramidal neurons especially in the mice injected with TauO (Figure 2c,d, white arrows). Interestingly, FK506 treatment not only mitigated the TauO-induced increase in CaN but also restored the levels to a comparable state observed in the control groups. Importantly, FK506 alone did not affect CaN levels, indicating that its impact is specifically related to TauO. These observations offer insights into the impact of TauO on CaN levels and distribution across various brain regions, underscoring the potential of FK506 in alleviating the effects induced by TauO.

### 2.2. FK506 Treatment Decreases Levels and Phosphorylation of ICV-Injected TauO

To assess the impact of FK506 treatment on the hippocampal levels of exogenously administered TauO, we conducted double immunofluorescence staining for total Tau and phosphorylated Tau (pTau) (Ser202/Thr205-AT8) as illustrated in Figure 3. The images show Tau (green) and pTau (red) immunoreactivity in hippocampal regions (DG, CA1, and CA3) of mice injected with TauO, while no immunoreactivity signal was detected in control mice (Figure 3). Interestingly, Tau and pTau were predominantly localized in the soma and in the nuclei of the granule and pyramidal neurons of the DG and CA1/CA3, respectively. Extensive quantitative analysis of immunoreacted sections revealed a significant decrease in Tau and pTau levels after FK506 treatment in the DG (Figure 3a–d) and CA3 (Figure 3i–l). Additionally, we observed a decreasing trend of Tau and pTau after FK506 treatment in the CA1 region (Figure 3e–h). Western blot analyses of TauO (T18) and pTau (AT8) levels across all experimental groups supported these findings (Appendix A).

Employing the in vivo fluorescence imaging procedure outlined in the methods, we longitudinally monitored TauO levels in mice ICV injected with pre-labeled TauO (Figure 4a–d). The fluorescence threshold, established by control mice baseline signals, guided the analysis of the region of interest (ROI) around the fluorescent signals. Remarkably, we observed a reduction in TauO fluorescence intensity starting at 3 h post-FK506 treatment, peaking at 6 h (Figure 4a,b). Imaging and analysis of the brains collected after 6 h post-FK506 revealed a significant decrease in the TauO signal in all FK506-treated mice (Figure 4c,d), aligning consistently with the above findings. Overall, these data suggest that CaN inhibition by FK506 facilitates the clearance of injected TauO.

### 2.3. FK506 Treatment Prevents the Formation of CaN-GSK3β Complex

The phosphorylation of Tau protein involves several kinases, and the specific kinase engaged depends on the phosphorylation site of the amino-acid residue [26]. GSK3β stands out as the most active and extensively studied kinase within this pathway, and CaN is recognized as the responsible phosphatase for dephosphorylating GSK3β [27]. This process results in the activation of GSK3β, which, in turn, phosphorylates Tau, contributing to the formation of the CaN-GSK3β-Tau complex. To confirm this association, double immunofluorescence staining of CaN and GSK3β was performed in the DG, CA1, and CA3 regions of mice from all experimental conditions (Appendix A). The images and quantitative analyses demonstrated elevated GSK3β levels in TauO-treated mice, a phenomenon reversed by FK506 treatment. Interestingly, Pearson’s correlation coefficient for colocalization analysis unveiled increased colocalization of CaN and GSK3β in TauO-treated mice, a phenomenon abolished by the FK506. To further verify this association, we conducted a qualitative proximity ligation assay (PLA) (Appendix A) to verify the direct interaction between CaN, GSK3β, and TauO. As showed in Appendix A, the presence of TauO produced positive PLA signals for CaN/GSK3β (upper panel), GSK3β/AT8 (central panel), and CaN/AT8 (lower panel) interactions. Notably, these positive PLA signals were no longer evident upon FK506 treatment.

### 2.4. FK506 Restores TauO-Induced Memory Deficits and Synaptic Alterations

The disruption of synaptic processes by TauO has been closely linked to cognitive decline in neurodegenerative diseases [10,14]. However, the connection between TauO-driven memory and synaptic dysfunctions and the dysregulation of CaN remains unexplored. To understand the involvement of CaN in TauO-induced neuronal toxicity and the physiological outcome of the observed CaN overexpression and hyper-activation, we investigated the effect of FK506 on TauO-induced memory deficits and synaptic alterations. Behavioral tests were conducted on mice injected with TauO with or without FK506 treatment, alongside control mice (Figure 5a). While no differences were observed during training (Figure 5b), TauO injection led to significant reductions of novel object recognition (NOR) performances both for short-term (2 h after training) and long-term memory (24 h after training) (Figure 5c,d). Object discrimination index (ODI) revealed significantly lower values in the TauO-treated group, indicating impaired hippocampal-dependent memory functions. Interestingly, administration of FK506 rescued ODI to levels comparable to those of the control groups in both short-term and long-term memory tests, suggesting a central role for CaN in TauO-induced memory impairment.

Given the known role of TauO in suppressing the expression of LTP and the important role played by CaN in proper synaptic processes [28], our goal was to elucidate the possible involvement of CaN-dependent pathways in TauO-induced suppression of LTP. Field-electrophysiology experiments revealed a significant decrease in the amplitude of field excitatory post-synaptic potentials (fEPSP) in the hippocampal region CA1 following high-frequency stimulation (HFS) of Schaffer Collaterals after TauO incubation, compared to the control (Figure 6a–d). Interestingly, FK506 treatment restored fEPSP to levels non-significantly different from controls. Details of input and output data are described in Appendix A.

Taken together, the behavioral and electrophysiological results suggest that TauO-induced synaptic toxicity relies on the suppression of LTP in a CaN-dependent fashion.

### 2.5. The Impact of Chronic FK506 on Neuropathology and Autophagy in 3xTgAD Mice

The findings described so far delineate the role of CaN on TauO-induced toxicity through its inhibition with FK506. To assess whether chronic FK506 treatment in a mouse model with established AD pathology would yield similar outcomes, we administered either FK506 or PBS via IP for 14-day period to 7-month-old 3xTgAD mice. Subsequently, we analyzed the levels of CaN, AβO, TauO, and the impact on the autophagic process. To validate the efficacy of the extended treatment with FK506, we evaluated hippocampal CaN levels by immunofluorescence staining. Chronic inhibition of CaN led to a significant decrease in CaN in DG, CA1, and CA3 as compared with control mice (Appendix A). Furthermore, immunofluorescence (Figure 7a–c) and Western blot (Figure 7d,e) analyses revealed notable reduction in AβO and TauO levels in the hippocampus of 3xTgAD subjected to chronic treatment with FK506. Recent evidence in human AD samples suggests that in AD the dysfunctional autophagy leads to the accumulation of TauO and insoluble Tau aggregates, a phenomenon that can be mitigated by autophagy restoration [29]. To assess the impact of CaN inhibition on the autophagic pathway, we analyzed the expression levels of proteins associated with autophagy in the hippocampus of 3xTgAD mice subjected to chronic FK506 treatment compared to untreated mice. Western blot analyses of autophagic markers indicated an overall trend toward restoration (Appendix A), aligning closely with the previous observations in humans [29]. Particularly, FK506 treatment resulted in a significant increase in Atg9, Atg16L1, and Beclin-1, with Atg5 and LC3B showing a clear upward trend, although not reaching statistical significance. Consistent with the above findings, these results underscore the role of CaN in TauO-induced toxicity through autophagy suppression, highlighting the potential therapeutic efficacy of CaN inhibition in addressing AD pathology, both acutely and chronically.

## 3. Materials and Methods 

### 3.1. Animals

*Acute treatment with FK506***.** To study the effect of CaN inhibition on TauO clearance, we used C57BL/6J male wild-type mice. At 12 weeks of age, mice were subject to intracerebroventricular (ICV) injection of either TauO or artificial cerebrospinal fluid (aCSF) and intraperitoneal (IP) administration of CaN activity inhibitor FK506 or phosphate buffer solution (PBS). Recombinant TauO were prepared and characterized following established and published protocols [24,30]. Mice were anesthetized by exposure to halothane vapors and ICV injected using the modified free-hand method previously described [16]. Briefly, a 29-gauge needle held with forceps to leave 4.5 mm of the needle tip exposed was lowered 1 mm posterior and 1 mm lateral of the bregma. The needle was connected to a 25 µL syringe using a 0.38 mm polyethylene tubing. Infusions were performed by an electronic programmable micro-infuser (Harvard Apparatus, Holliston, MA, USA), which was used to deliver 3 µL/mouse at a rate of 3 µL/min of either aCSF or 0.55 µM recombinant TauO in aCSF. After ICV injection, the needle was left in place for 2 min to assess proper administration while the mouse was allowed to recover lying on a heated pad under warm light. At 18 h after ICV, animals were given an IP injection of either 0.1 M PBS or 10 mg/Kg CaN inhibitor FK506 in PBS. The dosage of FK506 10 mg/kg was established based on previous studies demonstrating its efficacy and safety in similar models [31,32]. Six hours post-IP, mice were sacrificed, and brains collected for immunofluorescence or biochemical analyses. The animals were divided into four groups, according to the treatment received, both ICV and IP. These groups are referred to as: PBS + aCSF for animals receiving PBS IP and aCSF ICV; PBS + TauO for mice IP injected with PBS and ICV with TauO; FK506 + aCSF for animals receiving FK506 IP; and aCSF ICV; FK506 + TauO for mice IP injected with FK506 and ICV with TauO. Animal experiments were performed under a protocol approved by UTMB’s Institutional Animal Care and Use Committee, ensuring that the animals experienced the minimal amount of pain/discomfort. All animals were housed under USDA standards (12:12-h light/dark cycle, ad libitum food and water) at the UTMB vivarium. Graphical representation of the experimental design is shown in Figure 1.

### 3.2. In Vivo Imaging of the Effect of FK506 Treatment on Mice ICV Injected with Labeled TauO

To conduct this experiment, TauO were labeled by using the IVISense^TM^ 680 NHS Fluorescent Labeling Kit–VivoTag (Perkin Elmer–FP1473, Waltham, MA, USA; Lot#3065872) following the manufacturer’s instruction. Briefly, 500 µg of TauO were incubated with 50 µL of sodium bicarbonate and 2 µL of IVISense^TM^ 680 NHS Fluorescent Dye solution in the dark for 2 h using a Fisherbrand™ Mini Tube Rotator (cat# 88-861-051; Fisher Scientific, Hampton, NH, USA) rotating at a speed of 15 rpm. Afterward, the column provided in the kit was washed twice with PBS. The TauO-dye solution was loaded into the column and centrifuged at 1000× *g* for 2 min. The labeled TauO (TauO680) was collected as the flow-through and the degree of labeling (DOL) was calculated to be 2.3. Following the same experimental design described in the previous paragraph, C57BL/6J mice were ICV injected with either aCSF or TauO680 18 h prior to FK506 treatment delivered by IP injection. Images were acquired using the IVIS^®^ Spectrum In Vivo Imaging System (Perkin Elmer, Waltham, MA, USA). Mice were initially imaged immediately after TauO680 injection to confirm the presence of labeled oligomers. Subsequently, imaging sessions were conducted at 0–1–3–6 h post FK506 treatment to monitor TauO680 signal over time. The ROIs were selected and analyzed by using ImageJ software 1.54j (https://imagej.net/, NIH (accessed on 16 May 2024)).

### 3.3. Chronic Treatment of 3xTgAD Mice with FK506

Seven-month-old 3xTgAD male mice were chronically IP treated with either 0.1 M PBS or 1 mg/kg of FK506 every day for a total of 2 weeks. The FK506 dosage of 1 mg/kg was determined based on prior research highlighting its effectiveness and safety in comparable experimental models [33,34]. At the end of the treatment, mice were sacrificed and levels of CaN, AβO, TauO, and autophagy markers were evaluated by immunofluorescence and Western blot following the protocols described in the next paragraphs.

### 3.4. Immunofluorescence

According to IACUC-approved animal protocols and procedures, animals were deeply anesthetized with halothane vapors and perfused transcardially with 0.9% saline. Brains were carefully removed, post-fixed in 4% PFA for 48 h at 4 °C, and cryoprotected by suspension in 30% sucrose solution for 48 h at 4 °C. Brains were then embedded in O.C.T. compound (Tissue-Tek, Tokyo, Japan) and stored at −80 °C. For immunohistochemical experiments, mouse brain sections of 12 µm were cut using a cryostat (Microm HM 525, Thermo Fisher Scientific, Waltham, MA, USA) and collected onto Super Frost slides (catalog #12–550-15, Thermo Fisher scientific, Waltham, MA, USA). Slides were post-fixed in 4% paraformaldehyde for 30 min. After three washes in 0.1 M PBS, pH 7.4, non-specific binding sites were blocked with 5% bovine serum albumin (BSA, Sigma-Aldrich Inc., Saint Louis, MO, USA)/10% normal goat serum (NGS, Thermo Fisher Scientific, Waltham, MA, USA) and sections were permeabilized with 0.5% Triton X-100/0.05% Tween-20 for 1 h at room temperature (21 °C). Incubation with primary antibodies in 0.1 M PBS containing 1.5% NGS was carried out overnight at 4 °C. Slides were then washed three times in PBS and incubated for 1 h at room temperature with appropriate secondary antibodies diluted in 0.1 M PBS containing 1.5% NGS. Slides were again washed three times in 0.1 M PBS, rinsed in 70% ethanol and then incubated for 10 min at room temperature in 70% ethanol containing 0.3% Sudan Black to remove lipofuscin autofluorescence. Slides were briefly washed in distilled water before mounting with Fluoromount-G containing 4′,6-diamidino-2-phenylindole (DAPI) (cat# Cat. No.: 0100-01 SouthernBiotech, Birmingham, AL, USA) and sealed. All antibodies used are reported in Table 1.

All immunoreacted sections were acquired with a Keyence BZ-X800 (Keyence Corporation, Itasca, IL, USA) microscope, by using 60× and 100× immersion oil objectives. For all analyses, each animal was analyzed as previously described [29]. Briefly, an average of three sections were analyzed for each animal/area (CA1, CA3, DG) and images were taken at 1920 × 1440 pixel resolution, with a z-step size of 1 μm at 14 μm thickness. For the feasibility of the quantification, all layers from a single image stack were projected on a single slice (stack/Z projection). Quantitative analyses were performed using ImageJ software 1.54J (https://imagej.net/, NIH (accessed on 25 May 2024)). We analyzed the intensity of fluorescence for each marker per area (Integrated Density, IntDen) and the colocalization coefficient when colocalization between two markers was studied. Representative images were composed in an Adobe Photoshop CC2020 format.

### 3.5. Proximity Ligation Assay (PLA)

Proximity ligation assay (PLA) was performed on slides prepared according to the immunofluorescence paragraph for all experimental conditions. The Duolink^®^ In Situ Red Starter Kit Mouse/Rabbit (cat# DUO92101-1KT, Millipore-Sigma, St. Louis, Missouri, USA) was utilized, following the manufacturer’s instructions. This involved a blocking phase, incubation with primary antibodies, and subsequent incubation with probes recognizing the primary antibodies. The ligase-mediated circular DNA formation, amplified by DNA polymerase, emitted a fluorescent signal representing the interaction between the analyzed proteins. CaN/p-Tau/GSK3β interactions were assessed by PLA using the same antibodies and incubation conditions (concentration and time) as described in the immunofluorescence section. Sections with immunoreactivity were imaged using a Keyence BZ-X800 microscope (Keyence Corporation, Itasca, IL, USA) with a 20× objective.

### 3.6. Western Blot Analyses

For biochemical analyses, brain tissue was dissected into individual areas including hippocampus, frontal cortex, and parietal-occipital cortex, then immediately frozen at −80 °C. For Western blot analyses, the hippocampal cytosolic fractions from mice ICV injected with either aCSF or TauO and IP treated with either PBS or FK506 were processed by using the Nuclear/cytosolic Fractionation Kit (cat # K266-25; Biovision, Exton, PA, USA) according to the manufacturer’s instructions. Briefly, 20 mg of brain tissue were cut into small pieces and homogenized with cold PBS and centrifuged at 500× *g* for 3 min. The pellet was incubated for 10 min on ice with cytosol extraction buffer A (CEB-A) plus DTT and protease inhibitor. Next, cytosol extraction buffer B was added and incubated for 1 min following by a centrifugation at 16,000× *g* for 5 min. The supernatant, which represents the cytosolic fraction, was collected and quantified by using the Pierce™ BCA Protein Assay Kit (cat# 23227, Thermo Fisher Scientific, Waltham, MA, USA).

For Western blot analyses on total protein extracts, hippocampi were lysed with 1 × radioimmunoprecipitation assay buffer (RIPA buffer, cat# 89900; Thermo Fisher Scientific, Waltham, MA, USA) with 1% protease and phosphatase cocktail inhibitors for 30 min on ice. The protein content was quantified by using the Pierce™ BCA Protein Assay Kit (cat# 23227, Thermo Fisher Scientific, Waltham, MA, USA) and an equal amount was loaded into a 4–20% Mini-PROTEAN^®^ TGX™ Precast Protein Gels (cat #4561096; BioRad, Hercules, CA, USA) followed by 1 h transfer to Amersham Protran nitrocellulose transfer membranes (cat# 10600001; GE Healthcare-Life Sciences, Marlborough, MA, USA) at 95 V at 4 °C. The membranes were blocked using Odyssey blocking buffer (LI-COR, Lincoln, NE, USA) for 1 h at RT and incubated at 4 °C overnight with the primary antibodies (see Table 1 for details). All primary antibodies were prepared in a 1:1 solution of 1 × Tris-buffered saline solution with Tween (TBST) and Odyssey blocking buffer. After incubation, the membranes were washed three times with 1 × TBST (10 min each) and incubated 1 h with LI-COR secondary antibodies diluted at 1:10,000 in 1 × TBST-Odyssey blocking buffer at RT. The membranes were again washed three times for 10 min each. Western blots were imaged using an LI-COR.

Odyssey infrared imaging system, application software version 3.0.30. The density of immunoreactive bands were measured using ImageJ FIJI software 1.54J (https://imagej.net/, NIH, Bethesda, MD, USA (accessed on 21 May 2024)).

### 3.7. Field Excitatory Postsynaptic Potential Recordings

Our standard protocol was used as previously described [35,36,37]. Briefly, mice were deeply anesthetized with isoflurane and transcardially perfused with 25 to 30 mL of carbogenated (95% O_2_ and 5% CO_2_ gas mixture) N-methyl-D-gluconate-artificial cerebrospinal fluid (NMDG-aCSF) at room temperature. Transverse brain sections of 350 µm containing Schaffer collateral synapses were generated using the Compresstome VF-300 (Precisionary Instruments, Greenville, NC). An initial protective recovery was performed in the cutting solution at 32–34 °C for 12 min and then transferred to carbogen bubbling HEPES-aCSF recovery solution at room temperature. After recovery, slices were perfused in carbogen bubbling room temperature normal artificial cerebrospinal fluid (naCSF) at a rate of approximately 3 mL/min. In the recovery phase, slices were isolated to a separate chamber and incubated with 0.05 µM TauO for 45 min prior to recording. Slice treatment with CaN inhibitor FK506 was applied 15 min prior to recording and 30 min after TauO incubation. After the treatment, the slices were briefly washed by placing them in oligomer-free recovery artificial cerebrospinal fluid for 5 min before placing them on the recording stage. Slices were recorded in carbogenated standard recording naCSF. Evoked field excitatory post-synaptic potentials (fEPSPs) recordings were performed by stimulating the Schaffer collateral pathway (located in stratum radiatum) using a stimulating electrode of ~22 kΩ resistance placed in the CA3 region and glass recording electrodes in CA1 region. Current stimulation was delivered through a digital stimulus isolation amplifier (A.M.P.I, Jerusalem, Israel) and set to elicit a fEPSP approximately 30% of maximum for synaptic potentiation experiments using platinum–iridium tipped concentric bipolar stimulating electrodes (FHC Inc., Bowdoin, ME, USA). Using a horizontal P-97 Flaming/Brown Micropipette puller (Sutter Instruments, Novato, CA, USA), borosilicate glass capillaries were used to pull recording electrodes and filled with naCSF to obtain a resistance of 1–2 MΩ. Field potentials were recorded in CA1 stratum radiatum using an Ag/AgCl wire in CV7B headstage (Molecular Devices, Sunnyvale, CA, USA) located ~1–2 mm from the stimulating electrode. LTP was induced using a high frequency stimulation protocol (3 × 100 Hz, 20 s). Recordings were digitized with Digidata 1550B (Molecular Devices, Sunnyvale, CA, USA), amplified 100× and digitized at 6 kHz using an Axon MultiClamp 700B differential amplifier (Molecular Devices) and analyzed using Clampex 10.7 software (Molecular Devices). To assess basal synaptic strength, 250 μs stimulus pulses were given at 10 intensity levels (range, 100–1000 μA) at a rate of 0.1 Hz. Three field potentials at each level were averaged, and measurements of fiber volley (FV) amplitude (in millivolts) and fEPSP slope (millivolts per millisecond) were performed using Clampfit 10.7 software. Synaptic strength curves were constructed by plotting fEPSP slope values against FV amplitudes for each stimulus level. Baseline recordings were obtained for 10 min by delivering single pulse stimulations at 20 s intervals. All data are represented as a percentage change from the initial average baseline fEPSP slope obtained for the 10 min prior to HFS.

### 3.8. Behavioral Testing

Novel object recognition (NOR). NOR was performed as previously described [35,36,37]. Briefly, animals were habituated for two consecutive days in an open field box and assessed for normal locomotion and acclimation to the test environment. Twenty-four hours after the last habituation session, animals were subjected to training in a 10-min session of exposure to two identical, non-toxic objects (hard plastic items) in the open field box. The time spent exploring each object was recorded using an area of 2 cm^2^ surrounding the object and was defined such that nose entries within 2 cm of the object was recorded as time exploring the object. After the training session, the animal was returned to its home cage. After a retention interval of 2 h and subsequently 24 h, the animal was returned to the arena in which two objects, one identical to the familiar object but previously unused (to prevent olfactory cues and prevent the necessity to wash objects during experimentation) and one novel object. The animal explored for 10 min, during which the amount of time exploring each object was recorded. Objects were randomized and counterbalanced across animals. The animals were returned to their home cages with food/water ad libitum. The percentage of time exploring each object (familiar versus novel) is reported as an object discrimination index (ODI). An index above 0.5 is indicative of novelty associated with the object. Each mouse was tested at 2 h and at 24 h with the intention of assessing the shorter and longer time frames in memory recall. Different novel objects (color and shape) were used in the 24 h test compared to the 2 h test, to avoid performance deficits.

### 3.9. Statistical Analyses

Statistical analyses were performed using GraphPad Prism version 10.0.2 software. One-way ANOVA with post hoc corrections for multiple comparison (Tukey’s, Šídák’s, and Dunn’s) were used to detect significant differences between groups. For experiments on 3xTgAD mice, a one-tailed *t*-test was used to detect significant differences between groups. Data were expressed as mean ± SD or mean ± SEM. Significance was considered for * *p* < 0.05, ** *p* < 0.01, *** *p* < 0.001, **** *p* < 0.0001. (See figure legends for specifications).

## 4. Discussion

The presented study explores the intricate relationship between CaN and Tau pathology, shedding light on the potential therapeutic implications of CaN inhibition in AD. Our findings indicate that acute and chronic treatment with the CaN inhibitor FK506 exerts significant protective effects against toxicity across multiple aspects of AD pathology driven by both exogenously administered and endogenous TauO. Specifically, the acute treatment phase focused on the short-term impacts of FK506 on memory cognition and synaptic function, while the chronic phase investigated its long-term effects on autophagic processes. This dual approach provides a comprehensive understanding of FK506′s therapeutic potential in addressing both immediate and sustained aspects of Tau pathology.

CaN is a Ca^2+^/Calmodulin-dependent serine/threonine protein phosphatase particularly enriched in neurons where it regulates synaptic plasticity [38,39,40]. Structurally CaN is formed by a catalytic subunit A and a Ca^2+^ binding subunit B, it also includes an autoinhibitory domain (AI) that allows for inactivation of the protein in the absence of Ca^2+^ binding. Following Ca^2+^ binding, the autoinhibition is alleviated and CaN is activated [41,42,43]. Compelling evidence has established that small oligomeric forms of Aβ and Tau proteins are the most toxic species causing neuronal degeneration [44,45,46,47]. Even though a clear molecular mechanism for oligomer-driven toxicity is yet to be ultimately determined, previous studies demonstrated that oligomers majorly impact synaptic plasticity by altering Ca^2+^ homeostasis [48,49,50,51]. Thus, oligomer-induced increases in intracellular Ca^2+^ at the synaptic level might enhance the activation of CaN [52]. Previous studies demonstrated that injection with AβO and α-synO significantly increase CaN activity in the hippocampus [17,18,19]. Also, it was observed that CaN levels are altered in post-mortem AD brains [53]. Furthermore, the thigh regulation necessary for proper CaN functioning suggests that a dysregulation in CaN activity could be closely linked or even causative of memory dysfunctions, neuronal loss, and eventually cognitive impairment [54]. Currently, known CaN modulators and inhibitors attract considerable interest because of their potential use as correctors of pathological brain states [55]. Based on these premises, we hypothesized that TauO-driven synaptic impairment in AD could rely on abnormal CaN expression and activation. To test our hypothesis and provide novel insight to the intricate molecular mechanism of TauO-induced neurodegeneration, we thoroughly evaluated TauO impact on CaN expression levels in 12-week-old C57BL/6J male wild-type mice ICV injected with recombinant TauO, followed by treatment with the CaN activity inhibitor FK506. The investigation was focused on hippocampus due to its pivotal role in memory processing and early involvement in AD. Immunofluorescence analyses demonstrated a distinct increase in CaN fluorescence intensity levels in the DG, CA1, and CA3 regions of TauO-treated mice, particularly in neuronal cell soma. Remarkably, FK506 alone did not influence CaN signal, suggesting its neutral impact. The therapeutic potential of FK506 became evident as its administration successfully reversed the TauO-induced increase in CaN in the hippocampus, indicating a potential avenue for intervention in Tau-associated pathologies. A discrepancy was observed between the Western blot and immunofluorescence results, which may be attributed to differences in sensitivity and specificity between these techniques. While Western blot analysis of CaN protein levels did not reveal significant differences across experimental groups, the immunofluorescence data highlighted notable changes in CaN localization. This discrepancy may reflect the enhanced “visibility” of CaN in immunofluorescence experiments, potentially due to the aggregation of CaN with GSK3β, a known substrate for CaN, following Tau injection. This aggregation might not be adequately captured by Western blot analysis, which primarily measures overall protein levels rather than spatial localization or interactions. Thus, the nuanced nature of TauO-induced changes in CaN may be better assessed through immunofluorescence, emphasizing the importance of using multiple techniques to fully characterize the effects of TauO and FK506. Furthermore, to evaluate the possible beneficial effects of CaN activity inhibition, we explored the immediate effects of CaN inhibition on TauO, demonstrating a substantial reduction in TauO presence and phosphorylation following FK506 treatment. This effect was not only evident in the hippocampus but also extended to cortical regions, highlighting the broad impact of CaN inhibition on TauO throughout the brain. Moreover, our in vivo fluorescence imaging revealed a notable longitudinal decrease in TauO levels after FK506 administration, suggesting a clearance of TauO facilitated by CaN activity inhibition.

GSK-3β is reported as one of the most effective kinases in promoting Tau hyperphosphorylation and accumulation in AD [56]. One of the consequences of this phenomenon is the reduced interaction of Tau with microtubules, potentially leading to the destabilization of the microtubule network, thus leading to neuronal disfunction. Furthermore, phosphorylation of Tau by GSK3β or other kinases may result in several interactions with other proteins [57]. Interestingly, it has been shown that Tau can act upstream of GSK-3β by directly acetylating it. This inhibits GSK-3β ubiquitination and enhances its enzymatic activity, resulting in further hyperphosphorylated Tau [58]. These studies demonstrate a bidirectional relationship of GSK-3β with Tau [8]. CaN plays a pivotal role in this process by regulating the dephosphorylation and activation of GSK-3β at Ser-9, forming a stable complex with GSK-3β [27]. Consequently, increased CaN activity leads to a decrease in GSK-3β phosphorylation at Ser-9, concomitant with heightened GSK-3β activity. Overall, these findings underscore the formation of a stable and interactive complex involving CaN, GSK-3β, and Tau, playing a pivotal role in AD pathology. Based on this evidence, we evaluated the impact of CaN inhibition by FK506 on the formation of this critical complex in the hippocampus of wild type mice ICV injected with TauO. Our findings demonstrated that the injected TauO engaged with CaN, promoting the clustering of CaN and its association with GSK3β. Consequently, CaN dephosphorylated GSK3β, resulting in its activation which, in turn, leads to an increased TauO phosphorylation. The formation of the complex Tau-CaN-GSK3β was confirmed by PLA assay. Importantly, FK506 treatment disrupted this CaN-GSK3β-Tau complex, pointing towards a potential key mechanism through which CaN activity inhibition mitigates TauO-induced toxicity. Nevertheless, we do not exclude that the changes in the distribution of CaN upon TauO injection could involve the interaction of CaN with other factors such as NFAT or CREB [53].

FK506 selectively inhibits CaN hyperactivation, maintaining a basal physiological level of CaN function [55]. Thus, this highlights that selective CaN inhibition carried out by FK506 treatment has a potential restorative effect on TauO-driven toxicity and makes this FDA-approved immunosuppressant drug a valid candidate to counteract TauO-induced synaptic dysfunction at the basis of AD neuronal degeneration. Noteworthily, it was observed that patients chronically treated with FK506 to prevent solid organ transplant rejection show a significantly lower incidence of AD compared to the general population [22,23]. To further research the functional outcomes of acutely administered exogenous TauO and, more so, the role played by CaN in TauO-driven early cognitive impairments, we assessed memory functions following TauO and FK506 administration. Particularly, we demonstrated that recombinant TauO induced suppression of synaptic LTP and impairment of both short and long-term memory processes. More interestingly, FK506 treatment not only rescued memory deficits but also restored synaptic function, emphasizing the central role of CaN in TauO-induced neuronal toxicity. Although further measurements of synaptic indicators such as PSD-95 and synaptophysin will be important for illustrating synaptic repair and will be conducted in future studies, the LTP data described above suggest that FK506 may not only mitigate TauO-associated cognitive impairments but also contribute to the restoration of normal synaptic function, providing further evidence for the therapeutic potential of CaN inhibition in the context of AD pathology.

Tau oligomers are considered the most toxic species to neurons and are strongly implicated in the synaptic dysfunction underlying memory deficits in AD [5,58]. It is widely described that autophagy is the major cell mechanism responsible for removing protein aggregates, including TauO. However, in AD, this essential process becomes dysregulated [29,59]. In AD, the impairment of the autophagy-lysosome system affects the clearance of TauO, contributing to the accumulation of these toxic species and the progression of neurodegeneration. In recent findings, it has been highlighted that maintaining functional autophagy represents a crucial protective mechanism for preserving cognitive integrity in resilient individuals that remain non-demented despite the presence of substantial AD pathology [29]. In the present study, preliminary analyses of the involvement of autophagy in AD pathology revealed the restoration of TauO-induced suppression of autophagic markers by FK506 treatment. Although further investigation is necessary, the observed reinstatement of the autophagic pathway aligns with the recent evidence indicating the importance of autophagy in mitigating TauO-induced toxicity [29]. The triple transgenic (3xTg) AD mice are one of the most widely used AD models due to their unique ability to recapitulate both Aβ and Tau pathology [60] through the expression of APP695 (Swedish) and M146V PSEN1 transgenes combined with the expression of the human Tau MAPT (P301L) transgene [60,61]. Initial characterization of 3xTgAD mice demonstrated the presence of Aβ plaques at about 6 months of age and Tau tangles at 12 months of age, with synaptic dysfunction occurring before plaque and tangle deposition [61], with cognitive deficits observed at as early as 4 months of age [60,61,62]. To extend our understanding of the effect of CaN activity inhibition on Tau toxicity to a chronic system with endogenous oligomers, 7-month-old 3xTgAD mice were subjected to a 14-day treatment with FK506. The chronic inhibition of CaN activity resulted in reduced levels of CaN, AβO, and TauO, coupled with a restoration of autophagic markers. These findings not only provide valuable insights into the chronic effects of CaN activity inhibition but also suggest a potential long-term therapeutic strategy for mitigating AD neuropathology.

In conclusion, our study offers a comprehensive analysis of how inhibiting CaN activity affects TauO-induced synaptic impairment, addressing molecular, cellular, and behavioral dimensions. While further investigations are necessary to validate the translational potential of CaN activity inhibition in AD treatment, the observed consistent efficacy in both acute and chronic treatments underscores CaN as a promising therapeutic target. These findings highlight the need for continued research to fully understand the mechanistic aspects and to pave the way for potential interventions addressing various facets of AD pathology.

## Figures and Tables

**Figure 1 ijms-25-09092-f001:**
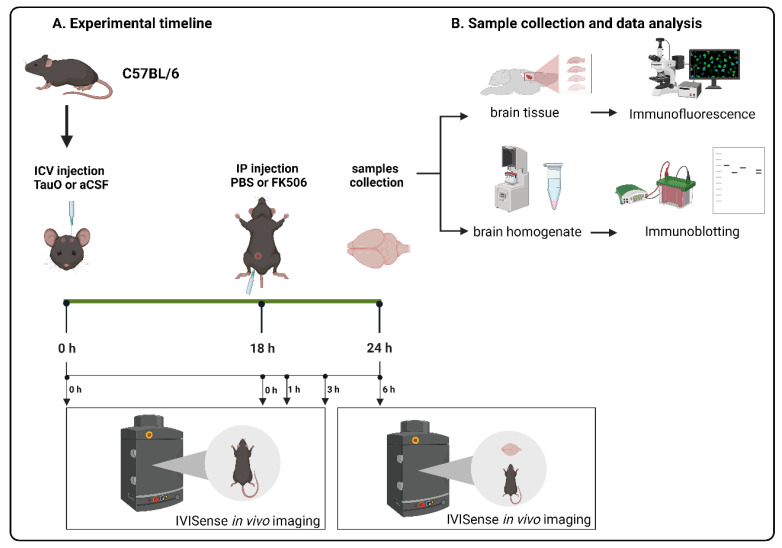
Graphical representation of experimental design.

**Figure 2 ijms-25-09092-f002:**
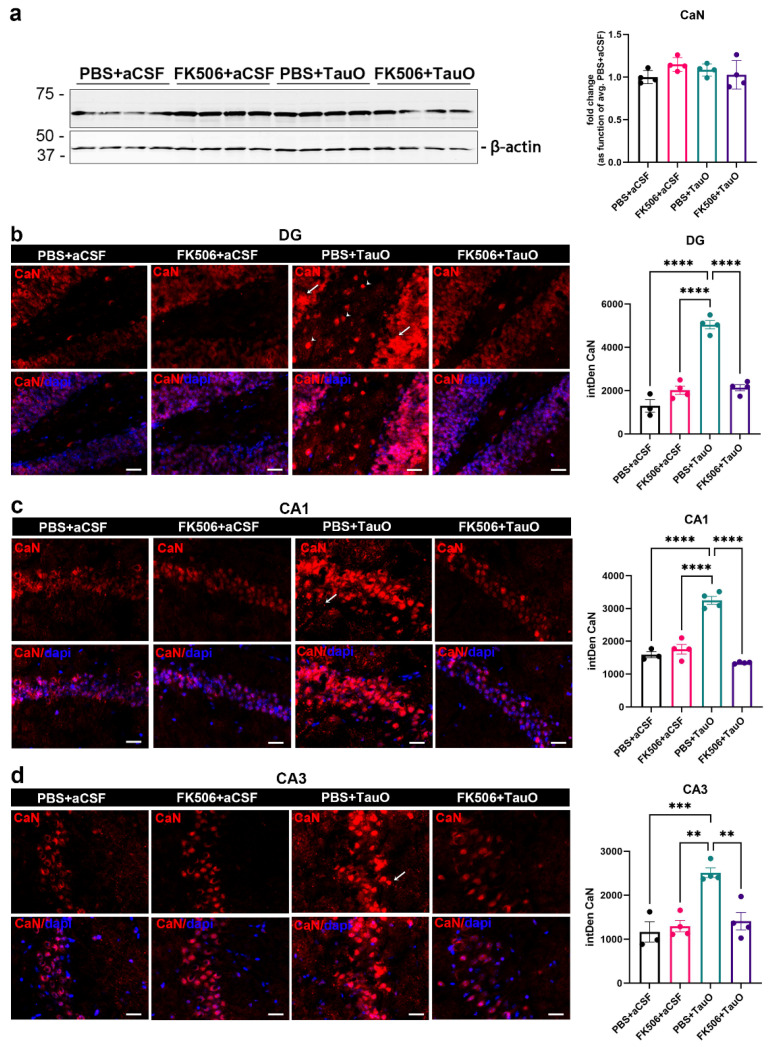
**Effects of FK506 administration on TauO-induced increased levels/visibility of CaN in the mouse hippocampus.** (**a**) Western blot analysis of CaN protein levels across the groups and relative quantification as function of average of PBS + aCSF mice; (**b**–**d**) (**Left** panels) Representative immunofluorescence images of CaN (red) levels in (**b**) dentate gyrus (DG), (**c**) CA1, and (**d**) CA3, from control mice (PBS + aCSF and FK506 + aCSF), mice ICV injected with TauO (PBS + TauO), and mice ICV injected with TauO and treated with FK506 (FK506 + TauO)). Original magnification (60×), scale bar (30 µm) and relative quantitative analyses (**right**) of the fluorescence intensity across all experimental groups. White arrows and white arrowheads indicate CaN distribution across the hippocampal regions analyzed. (**d**) Data represent the mean ± SEM; biological replicates n = 4 per group; one-way ANOVA with Tukey’s post hoc corrections for multiple comparison; ** *p* < 0.01, *** *p* < 0.001, and **** *p* < 0.0001.

**Figure 3 ijms-25-09092-f003:**
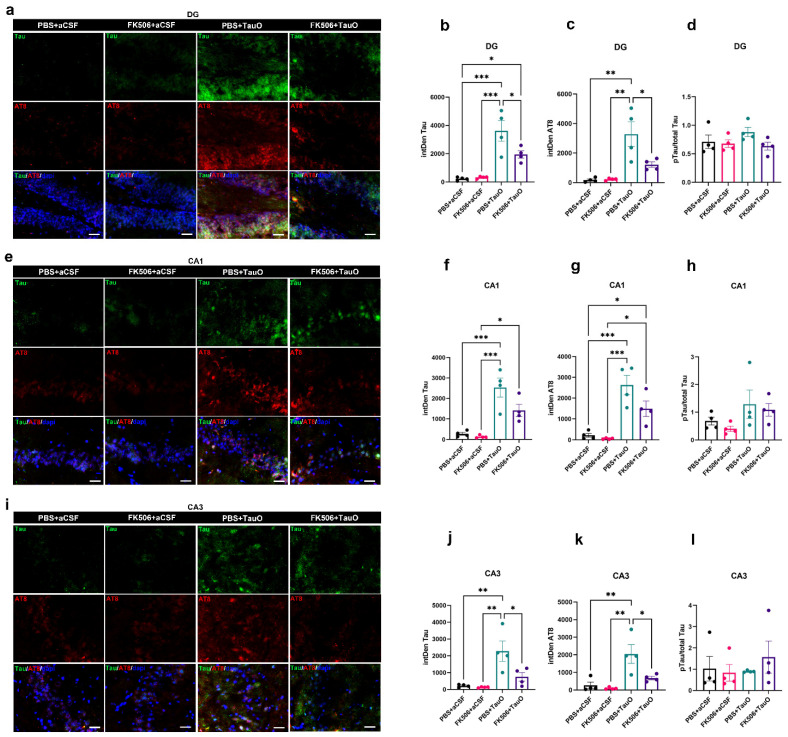
**Effects of FK506 administration on hippocampal TauO levels and phosphorylation.** Representative immunofluorescence images and relative quantification analyses of Tau (green), p-Tau (AT8-red), and their ratio in (**a**–**d**) dentate gyrus (DG), (**e**–**h**) CA1, and (**i**–**l**) CA3 of control mice (PBS + aCSF and FK506 + aCSF), mice ICV injected with TauO (PBS + TauO), and mice ICV injected with TauO and treated with Fk506 (FK506 + TauO). Original magnification (60×), scale bar (30 µm). Data represent the mean ± SEM; biological replicates n = 4 per group; one-way ANOVA with Tukey’s post hoc corrections for multiple comparison; * *p* < 0.05, ** *p* < 0.01, and *** *p* < 0.001.

**Figure 4 ijms-25-09092-f004:**
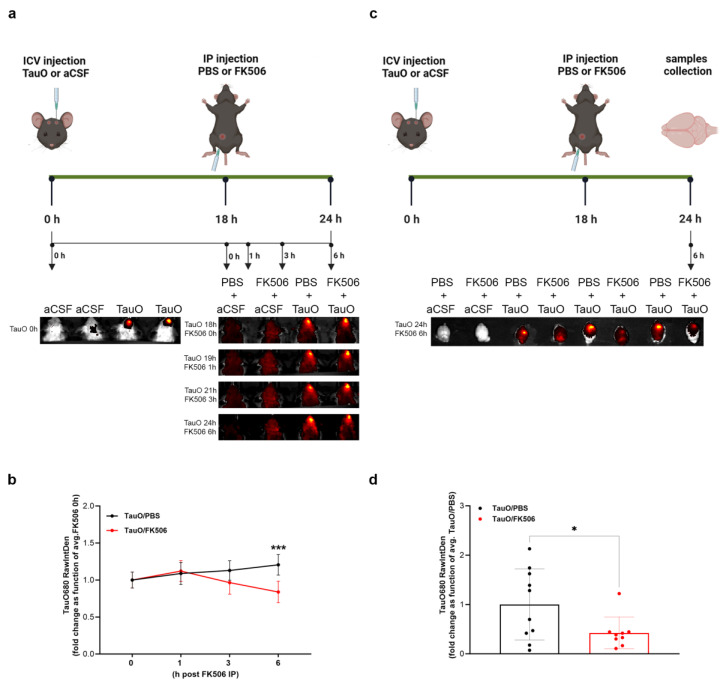
**Longitudinal in vivo imaging of mice ICV injected with labeled TauO and treated with FK506.** (**a**) Graphical description of the experimental design; (**b**) quantification analysis of the fluorescence intensity within the region of interest (ROI) in PBS + TauO and FK506 + TauO mice. (**c**) Graphical description of the experimental design with the collection of the brains as final step; (**d**) quantification analysis of the fluorescence intensity within the region of interest (ROI) in PBS + TauO and FK506 + TauO mouse brains. The fluorescence threshold, established by PBS + aCSF and FK506 + aCSF baseline signals, guided the analysis of ROI around the fluorescent signals. Data represent the mean ± SEM; t biological replicates n = 10 per group; one-way ANOVA with Šídák’s post hoc corrections for multiple comparison; * *p* < 0.05 and *** *p* < 0.001.

**Figure 5 ijms-25-09092-f005:**
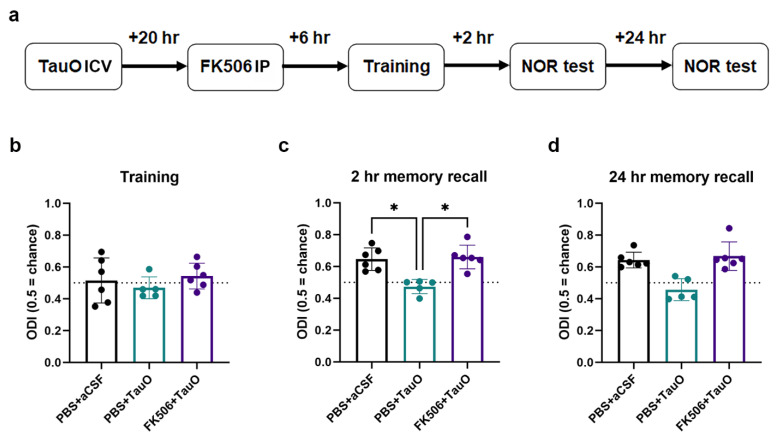
**TauO impairs hippocampal-dependent memory function in a CaN-dependent fashion.** (**a**) Graphical representation of novel object recognition (NOR) behavioral test performed on PBS + aCSF, PBS + TauO, and FK506 + TauO mice; time spent with old and new objects calculated as per object discrimination index (ODI) during (**b**) training, (**c**) 2 h recall, and (**d**) 24 h recall behavioral test phases. Data represent the mean ± SD; biological replicates n = 6 per group; one-way ANOVA with Dunn’s test for multiple comparison; * *p* < 0.05.

**Figure 6 ijms-25-09092-f006:**
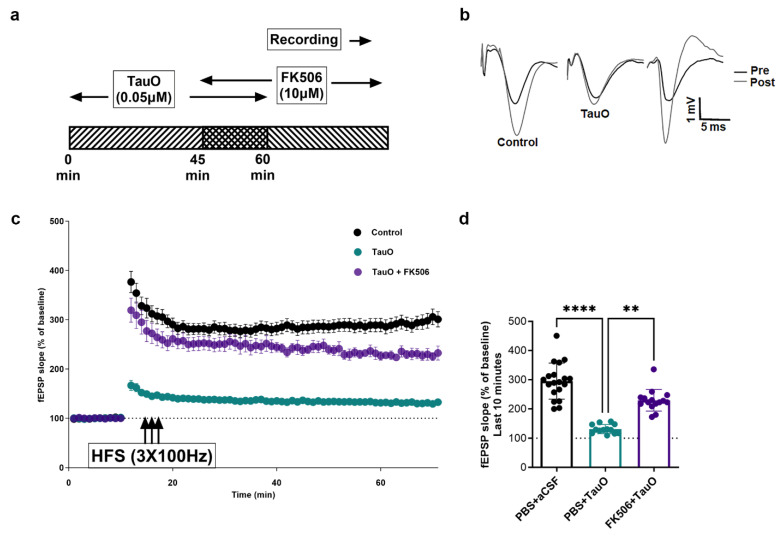
**TauO suppression of long-term potentiation in a CaN-dependent fashion.** (**a**) Graphical representation of experimental design; (**b**–**d**) high-frequency stimulation long-term potentiation in 6 control mice (PBS + aCSF), 6 mice ICV injected with TauO (PBS + TauO), and 8 mice ICV injected with TauO and treated with FK506 (FK506 + TauO). Data represent the mean ± SD; n = 6–8 animals were used for the treatment of the slices as shown in the schematic. The dots represent the total number of slices that were treated. n = 19 for control; n = 13 for TauO, and n = 15 for TauO + FK506. One-way ANOVA with Dunn’s test for multiple comparison; ** *p* < 0.01, and **** *p* < 0.0001.

**Figure 7 ijms-25-09092-f007:**
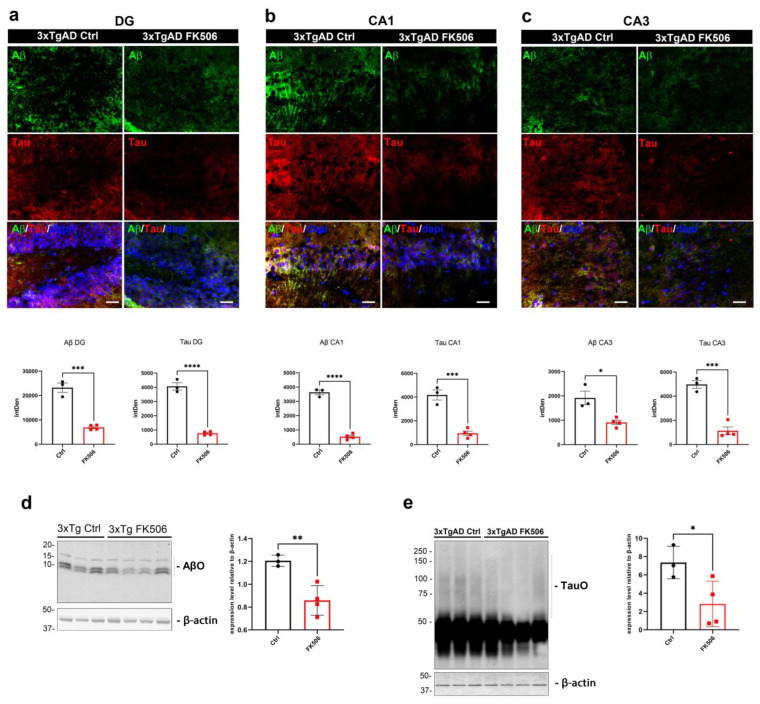
**Impact of chronic FK506 on AD neuropathology in 3xTgAD mice.** (**a**–**c**) Representative immunofluorescence images (upper panels) and relative quantification analyses (lower panels) of Aβ (green) and Tau (red) in (**a**) dentate gyrus (DG), (**b**) CA1, and (**c**) CA3 of control and FK506-treated 3xTgAD mice; (**d**,**e**) representative Western blot analyses of (**d**) AβO and (**e**) TauO and relative quantifications in the hippocampus of control and FK506-treated 3xTgAD mice. Original magnification (60×), scale bar (30 µm). Data represent the mean ± SD; biological replicates n = 3 (3xTgAD Ctrl) and n = 4 (3xTgAD + FK506); one-tailed *t*-test; * *p* < 0.05, ** *p* < 0.01, *** *p* < 0.001, and **** *p* < 0.0001.

**Table 1 ijms-25-09092-t001:** List of antibodies used for immunofluorescence and Western blotting.

Antibody	Source	Dilution	Ref.	Supplier	RRID	Application
Calcineurin	Rabbit polyclonal	1:300	ab3673	abcam, Cambridge, UK	AB_303991	IF; WB
Human Tau	Chicken	1:100	ab75714	abcam	AB_1310734	IF
pTau (AT8)	Mouse	1:100	MN1020	Thermo fisher	AB_223647	IF; WB
GSK3beta	Mouse	1:200	ab93926	abcam	AB_10563643	IF
IgG	Goat anti-rabbit Alexa Fluor 594	1:400	A-11012	Invitrogen, Carlsbad, CA, USA	AB_2534079	IF
IgG	Goat anti-mouse Alexa Fluor 488	1:400	A-11001	Invitrogen	AB_2534069	IF
IgG	Goat anti-mouse Alexa Fluor 594	1.400	A-11032	Invitrogen	AB_2534091	IF
IgY	Goat anti-chicken 488	1:400	A-11039	Invitrogen	AB_2534096	IF
IgY	Goat anti-chicken 594	1:400	A-11042	Invitrogen	AB_2534099	IF
T18	Rabbit polyclonal	1:1000	in-house			WB
β-actin	Rabbit polyclonal	1:5000	ab8227	abcam	AB_2305186	WB
P62	Rabbit monoclonal	1:1000	8025	Cell signaling, Danvers, MA, USA	AB_10859911	WB
LC3A/B	Rabbit monoclonal	1:1000	ab228525	abcam	AB_2827794	WB
Atg3	Rabbit polyclonal	1:1000	3415	Cell signaling	AB_2059244	WB
Atg5	Rabbit monoclonal	1:1000	Ab228525	abcam	AB_2650499	WB
Atg9	Rabbit monoclonal	1:1000	Ab228525	abcam	AB_10863880	WB
Atg16L1	Rabbit polyclonal	1:1000	Ab228525	abcam		WB
Beclin 1	Rabbit monoclonal	1:1000	Ab228525	abcam	AB_2692326	WB
6E10	Mouse monoclonal	1:1000	803002	Biolegend, San Diego, CA, USA	AB_2564654	WB

## Data Availability

Data related to the findings presented in this paper are available from the corresponding authors upon reasonable request.

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
