# Peer review of "Inhibition of Calcineurin with FK506 Reduces Tau Levels and Attenuates Synaptic Impairment Driven by Tau Oligomers in the Hippocampus of Male Mouse Models"

_ijms, 2024, doi:10.3390/ijms25169092_

Round 1

Reviewer 1 Report

Comments and Suggestions for Authors

The manuscript by Marcatti et al provides an important advancement from Taglialatela and colleagues regarding the pathophysiological role of calcineurin in Alzheimer’s disease. This work should be of great interest to the AD/ADRD research field, especially investigators who study Ca2+ signaling, tau mechanisms of pathology, and autophagy. Overall, the paper is clearly written and the results appear to be straightforward and interpreted appropriately. I really don't have any major comments or suggestions to offer. Congrats to the authors on a nice report.

Author Response

Thank you for your positive and encouraging feedback on our manuscript. We appreciate your recognition of the advancement our work represents in understanding the role of calcineurin in Alzheimer’s disease. We are pleased to hear that the paper is of interest to the AD/ADRD research field and that you found our results to be clear and appropriately interpreted. Your supportive comments and congratulations are greatly valued by our team.

Reviewer 2 Report

Comments and Suggestions for Authors

The authors construct acute and chronic treatment models and suggest that the inhibitor of calcineurin, FK506, effectively reduces TauO toxicity to attenuate synaptic impairment and promotes their clearance by enhancing the autophagy pathway. In general, this study is well-designed using a battery of histological, electrophysiological, and molecular biological techniques. The obtained data are original and interesting. However, some points need to be improved.

1. Lines 86 and 119: FK506 was used in one dose only in both acute and chronic experiments. Please explain how the concentration was determined or cite relevant references if possible.

 2. Line 360: The authors only used electrophysiological tests to illustrate that inhibition of calcineurin promotes synaptic function repair. This conclusion needed additional work to be fully substantiated by detecting relevant indicators of synaptic function, such as PSD-95 and synaptophysin. The authors should explain this limitation in the discussion.

 3. In the chronic treatment, the authors did not investigate the effects of FK506 on memory cognition and synaptic function, which were detected in the preceding acute treatment, rather choosing to focus on autophagic processes. Please explain the study design.

 4. The authors should explain the discrepancy between the results from the western blot and immunofluorescence shown in Fig. 2a and 2b-d.

 5. Line 264: Data were expressed as mean ± SD or mean± SEM. Please use one type of data analysis throughout the manuscript.

 6. Please add the name of the target protein to the right side of the band in Fig.2a.

 7. The quality of the immunofluorescence images needs to be enhanced.

 8. The overall level of the paper is good, but some of the details need to be taken care of. For example, "IF" is redundant (1:200 IF) in Table 1; Line 264: A full stop is missing between "group" and "Data"; Line 265 and 290: The p-value should be harmonized whether in uppercase or lowercase.

Comments on the Quality of English Language

The paper is generally satisfactory, but minor editing of English language is required.

Author Response

Thank you for your thorough review of our manuscript. We appreciate your constructive feedback and the opportunity to address the points you have raised. Below, we have outlined our responses to each of your comments:

  1. Lines 86 and 119: Determination of FK506 concentration. We acknowledge the necessity to justify the concentration of FK506 used in our experiments. The concentration was determined based on previous studies demonstrating its efficacy and safety in similar models. We have now included references to these studies in the materials and methods section of the revised manuscript to support the chosen of concentration.
  2. Line 360: Additional indicators of synaptic function. We agree that additional indicators of synaptic function, such as PSD-95 and synaptophysin, would provide a more comprehensive understanding of the effects of calcineurin inhibition. We have updated the discussion to address this limitation and suggest future studies incorporate these measurements for a more complete assessment of synaptic function.
  3. Chronic treatment study design. We designed our study with separate acute and chronic phases to explore different aspects of FK506’s effects on the impact of both exogenous and endogenous TauO. The acute phase focused on short-term impacts on memory cognition and synaptic function, which allowed us to evaluate immediate outcomes related to TauO toxicity. The chronic phase examined long-term effects, specifically on autophagic processes. This approach allowed us to investigate both immediate and sustained mechanisms of FK506’s supporting the therapeutic potential across different time scales and mechanisms. We have clarified the study design and rationale in the revised manuscript.
  4. Discrepancy between western blot and immunofluorescence results. The observed discrepancies between the western blot and immunofluorescence results may be due to differences in sensitivity and specificity between the two techniques. Furthermore, we believe that this enhanced “visibility” of CaN in immunofluorescence experiments may be due to the ability of CaN to aggregate with GSK3β after Tau injection, where the latter is the substrate for CaN. We expanded on this point in the discussion section.
  5. Consistency in data expression (Line 264). We apologize for the lack of clarity on this aspect. We have now standardized the expression of data as mean ± SEM throughout the manuscript to ensure consistency. The use of mean ± SD was a typo and has been confirmed through a thorough review of the raw data.
  6. Labeling of target Protein in Fig. 2a. We have added the name of the target protein to the right side of the band in Fig. 2a for clarity.
  7. Quality of immunofluorescence images. We have enhanced the quality of the immunofluorescence images to ensure they are clear and of high resolution, facilitating better interpretation of the results.
  8. Minor concerns. We corrected the redundancy of "IF" in Table 1, added the missing full stop between "group" and "Data" in line 264. We also have harmonized the p-value format throughout the manuscript, ensuring it is consistently presented.

We thank you once again for your valuable feedback and are confident that these revisions will significantly improve the quality and clarity of our manuscript.

Round 2

Reviewer 2 Report

Comments and Suggestions for Authors

The authors have satisfactorily addressed the comments in the revised manuscript. The manuscript has been substantially enhanced, and is now suitable for acceptance in the current version.